# COVID-19 and the Mental Health of Nursing Professionals in Brazil: Associations between Social and Clinical Contexts and Psychopathological Symptoms

**DOI:** 10.3390/ijerph191710766

**Published:** 2022-08-29

**Authors:** Maria do Perpétuo Socorro de Sousa Nóbrega, Moisés Kogien, Samira Reschetti Marcon, Angélica Martins de Souza Gonçalves, Marina Nolli Bittencourt, José Luís Cunha Pena, Maria Silvia Costa Silva, Darci Francisco Santos Junior, Dárcio Tadeu Mendes, Wanderson Carneiro Moreira, Suellen Cristina da Silva Chaves, Jheynny Sousa Alves, José Carlos da Silva Lins, Veônica de Medeiros Alves

**Affiliations:** 1School of Nursing, University of Sao Paulo, Sao Paulo 05403-000, Brazil; 2Faculty of Nursing, Federal University of Mato Grosso, Cuiabá 78068-600, Brazil; 3School of Nursing, Federal University of São Carlos, São Carlos 13565-905, Brazil; 4Department of Biological and Health Sciences, Federal University of Amapá, Macapá 68902-280, Brazil; 5School of Nursing, Federal University of Alagoas, Maceió 57051-090, Brazil

**Keywords:** COVID-19, mental health, nursing, anxiety, depression

## Abstract

The COVID-19 pandemic has had an important negative psychological impact on people worldwide, especially nursing professionals who seem to be more vulnerable to the development of psychopathological symptoms. **Objective:** To analyze relationships between variables from the social and clinical contexts with psychopathological symptoms in nursing professionals from different geographic regions of Brazil during the COVID-19 pandemic. **Methods:** Cross-sectional study carried out with 1737 nursing professionals from the five regions of Brazil. Data collection was carried out online with a questionnaire made available via Google Forms containing sociodemographic, occupational, and clinical questions, and the Symptom Assessment Scale-40-R, for the assessment of psychopathological symptoms. Differences between mean scores for the severity of psychopathological symptoms were assessed using the Mann–Whitney and Kruskall–Wallis tests. **Results:** There was a significant difference in the psychoticism domain scores according to Brazilian geographic region, with greater severity among professionals from the North and Northeast regions when compared with those from the South region. Social context variables (gender, age group, and marital status) and clinical variables (psychological and psychiatric follow-up; psychological or emotional support by the institution; family member, friend, neighbor, or co-worker with COVID-19, and death among them; use of psychiatric medication without a medical prescription; and taking steps to take care of their mental health) were significantly related to psychopathological symptoms. **Conclusions:** The results point to the importance of mental health promotion strategies for professionals through psychological or emotional support, as evidence indicates that this support can be a predictor of reduced psychological distress.

## 1. Introduction

In 2019, the first human cases of COVID-19 were identified in China. COVID-19 is a disease with high mortality, rapid progression and spread, and based on these aspects, the World Health Organization declared a state of crisis [1]. The high number of infected people manifesting serious forms of the disease with the threat of mortality overloaded health systems and services around the world, and directly impacted the work of health professionals, especially those working on the front line in the fight against COVID-19, generating a series of unique organizational, clinical and ethical challenges [2].

Many health professionals were affected by work overload due to the high number of contaminated patients requiring urgent specialized care, or had their routines drastically changed [2,3]. In the face of this, health care providers have adopted measures to avoid contagion of themselves and others in work settings in which adequate personal protective equipment was not always available [3].

Among the health professionals affected, nursing professionals stand out, representing, in quantitative terms, the largest contingent of workers in the fight against COVID-19 [4]. In pre-pandemic times, the literature already pointed to nursing professionals as a professional category highly exposed to work stress and mental illness [5,6], a situation that worsened during the pandemic [4,7].

Recent evidence indicates that nursing professionals who work in areas with high infection rates are among those with the most severe psychopathological symptoms among health professionals [8,9,10]. In addition, systematic studies on the psychological impact of the pandemic on the mental health of nurses point to a high prevalence of experiences of symptoms of anxiety, stress, depression, post-traumatic stress disorder and insomnia among nurses around the world [4,7]. Characteristics such as age, gender, occupation, specialization, type of activities performed and proximity to COVID-19 patients have shown a correlation with the severity of these psychopathological symptoms [11,12], not only among those who work in areas that directly serve patients with COVID-19, but also those who work in other areas [8,9,10].

It is important to emphasize that, despite the growing body of evidence available on the impact of the pandemic on the mental health of nursing professionals around the world, studies on Brazilian professionals, especially in the early stages of the pandemic, are still incipient [4,7]; moreover, the subject deserves attention, mainly because the South American continent drew the attention of the international community in the early stages of pandemic for the steep increase in the number of people affected by COVID-19 [13,14], especially in Brazil, the largest country in the region.

Brazil gained notoriety for the sharp increase in serious cases and deaths, which created a bottleneck in the capacity of its Health System, as well as for the differences in morbidity and mortality rates between the states of the federation. Updated data show that from the beginning of the pandemic until August 2022, 679,000 deaths from COVID-19 were recorded in Brazil, of which 173,000 were in the State of São Paulo and 7302 in the State of Piauí [15].

Thus, considering the scarcity of data assessing the mental health of Brazilian nursing professionals, the aim of this study is to analyze relationships between the social and clinical contexts and psychopathological symptoms in nursing professionals from Brazilian regions during the pandemic.

## 2. Materials and Methods

### 2.1. Study Design and Sample

This was an observational, analytical and cross-sectional study using an online-based survey. This report adhered to the Strengthening the Reporting of Observational Studies in Epidemiology (STROBE) guidelines. The research study took place between 22 April and 8 June 2020, during the critical period of the pandemic outbreak. The sample was formed by 1737 nurses, midwives, nursing assistants and technicians who work at any level of health care and/or work scenarios such as, for example, direct assistance, performing administrative/management functions, and/or in teaching and research in any of Brazil’s five geographic regions. A non-probabilistic sampling technique was used, with recruitment through the snowball method, in which the first respondents were asked to share the research link with other nursing professionals with whom they had contact.

### 2.2. Study Location

Brazil is the largest country in South America, located in the central-eastern part of the continent, with a territorial extension of 8,510,820,623 km^2^. The country is formed by 27 federative units, and it is divided into five main regions: North, Northeastern, Midwestern, Southeastern and South. Due to its immense geographic size, Brazil is a country of great geographic, social, economic and cultural contrasts [16], and these regional and intraregional inequalities affected the way the pandemic was experienced by each population group [17].

### 2.3. Data Collection and Variable Measurements

Data were virtually collected, through an electronic form built using the Google Forms^®^ tool, whose access link was available on social networks like Instagram^®^, Twitter^®^, Facebook^®^ and Whatsapp^®^ for 48 days. The e-questionnaire was divided into four sections. Section 1 contained sociodemographic and occupation data; Section 2 included questions about physical and mental health; Section 3 contained questions about the pandemic context and Section 4 highlighted psychopathological symptoms. The questions in the first three sections were designed by the researchers for use in the context of this study, while the assessment of psychopathological symptoms (Section 4) was based on the Symptoms Assessment Scale-40-R (SCL-40-R) [18].

The SCL-40-R is a self-report scale, composed of 40 questions that assess mental health based on the last 14 days. Multidimensional in nature, the instrument evaluates four groups of psychopathological symptoms: psychoticism, obsessive–compulsive, somatization and anxiety, each group of symptoms evaluated based on ten questions, with the choice of answers on a 3-point Likert scale measuring the intensity with which the respondent experiences each symptom (0 = none, 1 = a little, 2 = a lot). Each dimension was evaluated individually using a general mean score obtained from the sum of the items that compose it, divided by the number of items. Thus, each dimension presented a score that could range from 0 to 2 points, and the higher the score, the greater the severity of psychopathological symptoms. The scale was adapted from the Symptom Checklist-90-Revised (SCL-90-R) and validated for use in the Brazilian context, having shown good psychometric properties [18].

### 2.4. Data Analysis and Processing

Data were treated descriptively and inferentially using the Statistical Package for the Social Sciences (SPSS) software, version 26.0. In the descriptive analysis, continuous data were described through means and standard deviations, while categorical data were represented through their absolute and relative frequencies. The data did not show a normal distribution (Kolmogorov–Smirnov and Shapiro–Wilk tests with *p* < 0.05) and, therefore, for inferential analysis, the non-parametric Mann–Whitney and Kruska–Wallis tests were chosen. Dunn’s post-hoc test was used to verify differences between multiple groups when the Kruskal–Wallis test was significant. A significance level of 0.05 was adopted.

### 2.5. Ethical Approval

The National Research Ethics Committee approved the study, and it is in accordance with the Declaration of Helsinki of 1964. All participants signed an informed consent form.

## 3. Results

### 3.1. Sample Characteristics

The sample consisted of 1737, mostly female respondents (87.4%), the majority aged from 20 to 59 years (97.4%), married (53.4 %), and white (51.2%). Most did not initiate any psychological (92.3%) or psychiatric (94.0%) follow-up in the context of the pandemic. Most of them did not receive any psychological or emotional support from the institution where they work (78.0%). Most had had a family member, friend, neighbor or co-worker with COVID-19 (73.7%), and 27.4% experienced grief and took measures to take care of their mental health (47.7%). It was determined that 158 (9.1%) were using psychiatric medication without a medical prescription (Table 1).

### 3.2. Psychopathological Symptoms by Regions of the Country

Regarding psychopathological symptoms according to geographic region of the country, there was a significant difference only between the scores of the psychoticism domain (*p* = 0.002), and the post-hoc analysis showed that professionals from the North and Northeast regions presented greater severity of psychotic symptoms when compared with professionals from the southern regions of Brazil. No regional differences were found between obsessive–compulsive, somatic and anxiety symptoms scores (Table 2).

### 3.3. Psychopathological Symptoms, Social and Clinical Profile

The results showed that the means in all the investigated domains were higher among female professionals. There was a statistical significance in the psychoticism, obsessive–compulsive and somatization domains (Table 3).

The group aged 60 years or more had significantly lower means in all domains when compared to other age groups. Likewise, the highest means were found among single/widowed people, with statistical significance in psychosis, obsessive–compulsive and anxiety symptoms. When analyzing differences between age groups, it was observed that there is a difference in the anxiety domain between the age group of those under 19 years old and those over 60 years old (Table 3).

Psychoticism, obsessive–compulsive, somatization and anxiety symptoms were significant in relation to the following: psychological and psychiatric care; psychological or emotional support from the institution where they work; family member, friend, neighbor or co-worker with COVID-19, and death among them; use of psychiatric medication without a medical prescription; and taking steps to take care of their mental health (Table 3).

Higher psychopathological symptom intensity levels were identified among professionals who have undergone psychological and psychiatric care; those who had no psychological or emotional support from the institution where they work; those who have had a family member, friend, neighbor or co-worker with COVID-19, and death among them; those who have used psychiatric medication without a medical prescription; and those who have not taken steps to take care of their mental health (Table 3).

## 4. Discussion

There was predominance of females (87.4%) and those in the 20–59-year-old age range (97.4%) in the sample of this study. Nursing has cultural and historical-social aspects which characterize it as a care profession, seen as an eminently female attribution [19]. Furthermore, it is important to emphasize that these proportions are consistent with the findings of the Profile of Nursing in Brazil study, which interviewed 1,804,535 nursing professionals from all Brazilian regions and showed that 85.1% of the nursing workforce in the country was composed by women, with 97.1% of professionals aged under 60 years [20]. Regarding skin color, 51.2% of the participants in this study identified with white skin color, which is different from a study with Brazilian nursing students, where 54.5% claimed to have brown skin color and only 29.3% white [21].

In relation to the severity levels of psychopathological symptoms, it was found that the report of greater or lesser severity was associated with geographic, social and clinical characteristics of the sample. With the advance of the pandemic and the exponential growth in numbers of infected and dead people, it was necessary to restructure and reorganize spaces, routines and working hours to meet the urgent care demands imposed by the disease; nursing professionals, constituting the professional category who spend the most time with infected patients, were directly affected by work overload and high levels of work stress [22]. These elements, added to the specific fear of COVID-19, had a negative impact on the mental health and psychological well-being of these workers, producing a scenario of greater vulnerability to physical and mental fatigue, as well as for manifesting psychopathological symptoms [23].

Brazil is a country of great geographic size and with great sociocultural differences, and the COVID-19 pandemic did not develop equally in all regions at the same time. The states in the North and Northeast regions of the country were the most affected in the early stages, and had the highest incidence of coronavirus infection, as well as the highest mortality rates in the country, while the South region had considerably lower incidence and mortality rates in the same period [24].

These epidemiological differences in the non-uniform distribution of the disease caused professionals to experience its repercussions in specific ways, which may justify the self-report of greater severity of psychotic symptoms among professionals from the North and Northeast when compared with those from the South region. A study with Chinese health professionals showed that those who worked at the epicenter of the pandemic, in this case in Wuhan, were the most psychologically affected when compared to professionals from other less affected regions [25].

Another Chinese study showed that nurses experienced greater severity of psychopathological symptoms, including psychosis, during the first phases of coping with the COVID-19 pandemic [26]. Its greater magnitude among professionals in the regions which at the time of data collection experienced a collapse of the health system, and cemeteries with a demand for burials which exceeded the operational capacity of these spaces, can be explained by the daily experience of deaths in alarming numbers. Feelings of impotence, insecurity regarding the provision of adequate personal protective equipment, the lack of a sense of normality and work overload due to the unrestrained advance of the pandemic, are all elements associated with an increase in negative and hostile symptoms [27,28,29].

The results of this study also showed that gender, age and marital status were social determinants associated with the perception of the severity of psychopathological symptoms. The differences between mean scores according to these independent factors can generally be explained by the way these individual characteristics expose workers to allostatic overload [30], arising from the experience of common stressors in the profession, added to the new specific stressors which emerged in the pandemic context [31].

Regarding gender, women had higher means for psychosis, obsessive–compulsive and somatization symptoms than men. Although there is no consensus on differences in gender and magnitude of psychopathological symptoms in the literature, there is a tendency in the available evidence to demonstrate that women experience and report greater intensity of psychopathological symptoms, including in the pandemic context [32,33].

Men and women are affected by mental health problems; however, some of these are more prevalent among women, mainly because they are exposed to specific risk factors which attribute a greater chance of illness to them. In a study which related common mental disorders with sociodemographic variables and occupational stressors, it was found that the complexity and diversity of tasks for women result in a feeling of overload, as is the case of nursing women who are exposed to long working hours and an accumulation of domestic activities in addition to the roles of mother and wife [34].

The apparent amortization effect of age on the deleterious effects of the COVID-19 pandemic on people’s mental health has been observed in different studies, which corroborates the idea of lower severity of anxiety symptoms demonstrated in professionals aged over 60 years, even though they constitute an at-risk age group for worsening and mortality from COVID-19 [35,36,37].

Such findings may be related to the fact that older people are more adept at regulating their own emotions and managing everyday stress, making them less susceptible to pandemic concerns and less anxious about the risks to which they are vulnerable [37]. In the case of older health professionals, it is known that they generally have more years of personal and professional experience, constituting elements associated with greater resilience which can help mitigate the fear of COVID-19, and the anxiety symptoms associated with this event [38].

Regarding marital status, single or widowed professionals reported greater severity of psychosis, obsessive–compulsive and anxiety symptoms when compared to married professionals or those in a stable relationship. Marital relationships have recognized beneficial and protective effects for people’s mental health, but the ways in which they act salutogenically are varied. The role of companionship is highlighted as a source of social and emotional support which alleviates negative symptoms and feelings related to stressful situations in life, and concomitantly enhances feelings of happiness, satisfaction and well-being [39,40].

Marital/family relationships may be associated with mitigating the severity of psychopathological symptoms in this sample; however, the existence of a family structure or social nucleus with close relationships played an ambiguous role during the COVID-19 pandemic, exacerbating symptoms of psychosis, obsession–compulsion, somatization and anxiety. Nursing professionals who reported having someone close to them who was infected or deceased as a result of COVID-19 had higher scores in the four domains of the SCL-40-R.

The fear of one’s own death and/or of those close to oneself can constitute an anxiogenic experience and trigger mental suffering. On the other hand, family support can be important, with significant potential to alleviate the deleterious effects of work stress and mental distress [41]. The pandemic context, the fear of being infected and transmitting the disease to relatives/people close to them can be a stressful experience and is associated with higher levels of psychological damage.

Further evidence pointed out in this study refers to the fact that professionals who declared that they did not receive psychological/emotional support from the institution where they work had higher scores for the severity of psychopathological symptoms in the four evaluated domains. This finding is quite similar to what is currently experienced, and it even enables reflecting on the conditions of psychological preparation of these professionals for large-scale emergencies, such as the case of COVID-19 [42].

It should also be considered that given the pandemic we are experiencing, nursing professionals are part of one of the groups most exposed to contagion and emotional pain, which considerably affects mental health [43]. Therefore, spaces for intervention can be created in which nurses, as professionals with interpersonal communication skills, may listen to and support their peers [38,44].

A smaller portion of participants in this study (47.7%) reported that they take/took some measures to care for their own mental health during the pandemic, and they reported lower psychosis, obsessive–compulsive and somatization symptoms when compared with professionals who did not take/have not taken any mental health self-care measures.

Self-care can be a difficult attitude for nursing professionals, mainly because during the training process they are taught to focus on the care of others [45]. When facing a pandemic crisis, it is common for health professionals to focus their priorities on biological risk, pathophysiology and epidemiology to contain the disease, with the inevitable psychological repercussions being generally underestimated [1]. This dynamic is repeated especially in nursing, as it is a category of workers who spend most of their time alongside patients [46]. A scoping review emphasizes the need for health institutions to adopt protective, safety and support actions and psychosocial support in a short period of time [47].

In this study, professionals who reported the use of psychiatric medications without a medical prescription showed greater severity of all evaluated psychopathological symptoms. Self-medication is a common phenomenon in several regions of the world, including Brazil, and it seems to have increased its prevalence during the pandemic, especially among health professionals [48,49,50].

This study has some limitations. First, the cross-sectional observational design adopted makes it impossible to establish cause-and-effect relationships between the variables. Furthermore, it has an unavoidable volunteer bias due to the online collection method and snowball sampling method, both of which increase the risk of biased results. Finally, we could not control for every possible social or professional characteristic, and the observational essence of this study leaves the possibility of residual confounding.

## 5. Conclusions

Symptoms of psychosis were more common and significant among nurses in the North region. It was observed that male nurses, those older than 60 and married had the lowest means on the SCL-40-R; and these means were statistically significant in three of the four domains.

The results were significant and had lower means among those who underwent psychological and psychiatric care, had psychological or emotional support from the institution where they work, used psychiatric medication without a medical prescription, who took self-care measures for their mental health, and those who had no family member, friend, neighbor or co-worker who had died of COVID-19.

These results point to the need for promoting the mental health of nursing professionals in the context of health crises, and can serve as a reference for institutions and health systems to prepare themselves organizationally and in terms of assistance to face potential calamitous epidemiological scenarios similar to the one being currently experienced. The need for investments and the adoption of measures to promote the mental health of nursing professionals is reinforced, given that, both in the face of pandemics and in previous contexts, this professional category shows evidence of greater vulnerability to mental illness. Moreover, it is suggested that other studies should be conducted that explore the subjective aspects of the person/professional resulting in psychopathological symptoms.

## Figures and Tables

**Table 1 ijerph-19-10766-t001:** Social and professional profile of the sample of nursing professionals (n = 1737).

	N (%)	95% CI
Social profile		
Gender		
Female	1518 (87.4)	(85.8–88.9)
Male	219 (12.6)	(11.1–14.2)
Age range		
≤19 years	4 (0.2)	(0.1–0.5)
20–59 years	1692 (97.4)	(96.6–98.1)
≥60 years	41 (2.4)	(1.7–3.2)
Marital status		
Married/Stable union	927 (53.4)	(51.0–55.7)
Single/Widowed	810 (46.6)	(44.3–49.0)
Race/Skin Color		
White	889 (51.2)	(48.8–53.5)
Brown	635 (36.6)	(34.3–38.8)
Black	170 (9.8)	(8.5–11.3)
Others	43 (2.5)	(1.8–3.3)
In which region do you live in Brazil?		
Midwest region	283 (16.3)	(14.6–18.1)
Northeast region	416 (23.9)	(22.0–26.0)
North region	268 (15.4)	(13.8–17.2)
Southeast region	552 (31.8)	(29.6–34.0)
South region	218 (12.6)	(11.1–14.2)
Professional profile		
Professional performance		
Nurse	1466 (84.4)	(82.6–86.0)
Nursing Technician	230 (13.2)	(11.7–14.9)
Nursing assistant	37 (2.1)	(1.5–2.9)
Midwife	4 (0.2)	(0.1–0.5)
Weekly workload *		
20 h/week	73 (4.6)	(3.7–5.7)
36 h/week	382 (24.2)	(22.1–26.4)
40 h/week	643 (40.7)	(38.3–43.2)
44 h/week	136 (8.6)	(7.3–10.1)
More than 44 h/week	344 (21.8)	(19.8–23.9)
Total monthly income ** (Adding up all income sources)		
<1 MS	67 (3.9)	(3.0–4.8)
1–3 MS	707 (40.7)	(38.4–43.0)
4–6 MS	577 (33.2)	(31.0–35.5)
7–9 MS	252 (14.5)	(12.9–16.2)
≥10 MS	134 (7.7)	(6.5–9.0)
Are you currently working in direct care?		
No	675 (38.9)	(36.6–41.2)
Yes	1062 (61.1)	(58.8–63.4)

* The categories for the variable “Weekly workload” represent the standard weekly working hours in Brazil and respondents were instructed to choose the one that best represents their weekly workload. ** Current minimum monthly salary (MS) BRL 1045.00 = USD 206.25.

**Table 2 ijerph-19-10766-t002:** Comparison between psychopathological symptoms by regions of Brazil (*n* = 1737).

	Midwest Region	Northeast Region	North Region	Southeast Region	South Region	
	Mean ± SD	Mean ± SD	Mean ± SD	Mean ± SD	Mean ± SD	*p*-Value
Psychoticism	0.63 ± 0.49	0.66 ± 0.48	0.73 ± 0.50	0.63 ± 0.48	0.55 ± 0.45	0.002
Obsessive–compulsive	0.68 ± 0.52	0.73 ± 0.51	0.74 ± 0.52	0.68 ± 0.48	0.65 ± 0.51	0.123
Somatization	0.71 ± 0.54	0.74 ± 0.52	0.75 ± 0.55	0.71 ± 0.48	0.64 ± 0.53	0.056
Anxiety	0.44 ± 0.48	0.48 ± 0.50	0.47 ± 0.53	0.44 ± 0.48	0.41 ± 0.47	0.625

**Table 3 ijerph-19-10766-t003:** Comparison between social and clinical profiles and psychopathological symptoms (*n* = 1737).

	Psychoticism		Obsessive–Compulsive		Somatization		Anxiety	
	Mean ± SD	*p*-Value	Mean ± SD	*p*-Value	Mean ± SD	*p*-Value	Mean ± SD	*p*-Value
Social profile								
Gender		0.001		0.001		0.003		0.154
Female	0.66 ± 0.48		0.71 ± 0.50		0.72 ± 0.53		0.46 ± 0.48	
Male	0.55 ± 0.49		0.60 ± 0.51		0.62 ± 0.55		0.43 ± 0.49	
Age range		0.002		0.001		<0.001		0.002
≤19 years	0.53 ± 0.19		0.58 ± 0.30		0.83 ± 0.35		0.75 ± 0.19	
20–59 years	0.65 ± 0.48		0.71 ± 0.51		0.72 ± 0.53		0.46 ± 0.48	
≥60 years	0.38 ± 0.43		0.40 ± 0.43		0.30 ± 0.40		0.20 ± 0.43	
Marital status		0.010		0.049		0.052		<0.001
Married/Stable union	0.61 ± 0.47		0.68 ± 0.51		0.69 ± 0.54		0.42 ± 0.47	
Single/Widowed	0.67 ± 0.49		0.72 ± 0.50		0.73 ± 0.52		0.49 ± 0.49	
Race/Skin Color		0.110		0.280		0.490		0.783
White	0.61 ± 0.48		0.68 ± 0.50		0.69 ± 0.53		0.44 ± 0.48	
Brown	0.67 ± 0.48		0.71 ± 0.51		0.70 ± 0.53		0.46 ± 0.48	
Black	0.69 ± 0.53		0.76 ± 0.55		0.77 ± 0.58		0.50 ± 0.53	
Others	0.69 ± 0.53		0.74 ± 0.51		0.76 ± 0.53		0.51 ± 0.53	
Clinical profile								
Have you started some kind of psychological follow-up in the context of the COVID-19 pandemic?		<0.001		<0.001		<0.001		<0.001
Yes	0.97 ± 0.50		1.01 ± 0.51		1.06 ± 0.54		0.79 ± 0.50	
No	0.62 ± 0.47		0.67 ± 0.50		0.68 ± 0.53		0.42 ± 0.47	
Have you started some kind of psychiatric follow-up in the context of the COVID-19 pandemic?		<0.001		<0.001		<0.001		<0.001
Yes	1.01 ± 0.53		1.03 ± 0.53		1.05 ± 0.55		0.78 ± 0.53	
No	0.62 ± 0.47		0.68 ± 0.50		0.68 ± 0.53		0.43 ± 0.47	
Have you been using some psychiatric medication without a medical prescription in the context of the COVID-19 pandemic?		<0.001		<0.001		<0.001		<0.001
Yes	0.93 ± 0.48		1.01 ± 0.50		1.04 ± 0.50		0.80 ± 0.48	
No	0.61 ± 0.47		0.67 ± 0.50		0.67 ± 0.53		0.42 ± 0.47	
Have you received psychological/emotional support from the institution where you work/study in the context of COVID-19?		<0.001		<0.001		<0.001		<0.001
Yes	0.54 ± 0.45		0.60 ± 0.49		0.62 ± 0.52		0.38 ± 0.45	
No	0.67 ± 0.49		0.72 ± 0.51		0.73 ± 0.54		0.47 ± 0.49	
Do you have a family member, friend, neighbor, or colleague who was infected by the COVID-19 virus?		<0.001		<0.001		<0.001		<0.001
Yes	0.68 ± 0.49		0.74 ± 0.51		0.75 ± 0.54		0.48 ± 0.49	
No	0.53 ± 0.45		0.60 ± 0.49		0.59 ± 0.52		0.38 ± 0.45	
Have you had a family member, friend, neighbor, or colleague die from COVID-19?		<0.001		<0.001		<0.001		<0.006
Yes	0.74 ± 0.50		0.81 ± 0.54		0.80 ± 0.55		0.52 ± 0.50	
No	0.60 ± 0.47		0.65 ± 0.49		0.67 ± 0.53		0.42 ± 0.47	
Have you taken action(s) to take care of your mental health in the context of the COVID-19 pandemic?		0.043		0.035		0.023		0.097
Yes	0.62 ± 0.48		0.67 ± 0.50		0.68 ± 0.53		0.43 ± 0.48	
No	0.66 ± 0.48		0.72 ± 0.51		0.73 ± 0.54		0.48 ± 0.48	

## Data Availability

On demand to the corresponding author.

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
