# Peer review of "COVID-19 and the Mental Health of Nursing Professionals in Brazil: Associations between Social and Clinical Contexts and Psychopathological Symptoms"

_ijerph, 2022, doi:10.3390/ijerph191710766_

Round 1
Reviewer 1 Report
Dear Authors,
Unfortunately, your paper entitled “COVID-19 and the mental health of nursing professionals in Brazil: The correlation between social and clinical contexts and 3 psychopathological symptoms” can’t be considered, in my opinion, for publication in IJERPH, as the level of the scientific presentation is not adequate for this journal, the methods and the results are poorly provided. I left for you some comments, listed here below for the different sections of the manuscript.
Best regards,
the Reviewer.
General observation:
1. Why did you not cite the papers inside the text according to the journal’s requirements? (numbers inside squared brackets). Same for the reporting of the references in the ref. list at the end of the paper.
2. The language style used for the article is not adequate, nor for the English language, nor for the construction of the sentences based on the requirements of scientific writing.
Abstract
3. Please fully re-write the abstract: it is too long and completely not clear: it is not possible to understand what was done in your study from the abstract. It should be a summary of your study: objective (1-2 sentences), methods (2-3 sentences), results (2-3 sentences) and conclusions (1-2 sentences).
4. Line 13: please add “aim of the study is” before “to correlate”.
5. Line 14. Why “Cross-sectional study” with no sentence?
Introduction
6. Lines 37-38: data are too outdated, referring to more than 1 year ago.
7. Lines 42-43: the factors you are mentioning are not risk factors, but only associated factprs indicated in observational studies. Please find a systematic review on this topic and cite the data reported in the systematic review.
8. Lines 45-46: what is the reference for this statement?
9. Lines 47-50: too generic data, no numbers.
Material and Methods
10. Please re-write paragraph 2.1. Study design and sample adding verbs and subjects, as it seems there are not complete sentences here.
11. Line 68: what social networks? How? In what occasions?
12. Lines 70-72: what is the period you are referring to?
Results
13. Line 92. Why only for “white” it s specified “self-reported”? Also the other data are self-reported.
14. Line 99. Why “Table 01” and not Table 1?
15. Table 1. Not sure whether skin colors are the same as “ethnicity”.
16. Table 1. The regions f your country has to be better defined: this is not completely not clear for non-Brazilian readers.
17. Table 1. I can’t understand the subdivisions of Professional profiles: why it is based on yes/no responses? Why wasn’t all the population described according to the job activity? E.g. 79% nurse, 20% technician, 10% midwife, etc?
18. Table 1. “Weekly workload”: how these precise categories could be created if not included in the questionnaire? Or were the categories already defined in the questionnaire? Otherwise, I would have expected to have e.g. also someone with e.g. 32 or 38 hours/week
19. Table 1: “Total monthly income”: and what about those with 10 MS/month?
20. Table 02 and 03. Why 02 and 03 and not 2 and 3? Why “Source: The Author” in the footnote?
21. Table 3. What is “a”? “ab”?
22. Table 4. Formatting of the table issues.
Discussion
23. Limitations section has to be expanded.
Ref. lists
24. Not in line with journal style.
Author Response
We, the authors of the article entitled “COVID-19 and the mental health of nursing professionals in Brazil: relationships between social and clinical contexts and psychopathological symptoms”, are grateful for the comments made on our article. We consider all the comments and suggestions of the referees and present our response to each one of them below. We reinforce that all suggestions were accepted (as highlighted below). Changes made to the text are highlighted in yellow.
Reviewer 1
(x) Extensive editing of English language and style required
Answer: Provided the complete revision of the text for the English language.
- Why did you not cite the papers inside the text according to the journal’s requirements? (numbers inside squared brackets). Same for the reporting of the references in the ref. list at the end of the paper
Answer: All citations have been reviewed and adapted to the journal's recommendations, as indicated in the “Guidelines for Authors” section.- site do International Journal of Environmental Research and Public Health
- The language style used for the article is not adequate, nor for the English language, nor for the construction of the sentences based on the requirements of scientific writing.
Answer: In order to solve the problem mentioned, we provide a complete revision of the text for the English language..
Abstract
- Please fully re-write the abstract: it is too long and completely not clear: it is not possible to understand what was done in your study from the abstract. It should be a summary of your study: objective (1-2 sentences), methods (2-3 sentences), results (2-3 sentences) and conclusions (1-2 sentences).
Answer: Abstract has been completely restructured, presenting synthetically all the necessary elements.
- Line 13: please add “aim of the study is” before “to correlate”.
Answer: Recommendation met
- Line 14. Why “Cross-sectional study” with no sentence?
Answer: Changed the passage to increase the coherence of the sentence.
Introduction
- Lines 37-38: data are too outdated, referring to more than 1 year ago.
Answer: Data updated with current reference insert.
- Lines 42-43: the factors you are mentioning are not risk factors, but only associated factprs indicated in observational studies. Please find a systematic review on this topic and cite the data reported in the systematic review
Answer: Inserted citations from systematic reviews on risk factors associated with the mental health of nursing professionals during the COVID-19 pandemic.8. Lines 45-46: what is the reference for this statement?
Answer: Made changes to the introduction and provided a reference that substantiates the claim.9. Lines 47-50: too generic data, no numbers.
Answer: Made changes to the intro text.
Material and Methods
- Please re-write paragraph 2.1. Study design and sample adding verbs and subjects, as it seems there are not complete sentences here. -
Answer: Paragraph has been rewritten in an attempt to add more information for completeness of the section.
- Line 68: what social networks? How? In what occasions?
Answer: Entered information and rearranged text.
- Lines 70-72: what is the period you are referring to?
Answer: Entered period
Results
- Line 92. Why only for “white” it s specified “self-reported”? Also the other data are self-reported.
Answer: Fixed in text
- Line 99. Why “Table 01” and not Table 1?
Answer: adjusted
- Table 1. Not sure whether skin colors are the same as “ethnicity”.
Answer: Fixed and replaced by race/skin color as used in other publications of the International Journal of Environmental Research and Public Health
- Table 1. The regions f your country has to be better defined: this is not completely not clear for non-Brazilian readers.
Answer: Inserted in the text information about the organization and geographic division of Brazil.
- Table 1. I can’t understand the subdivisions of Professional profiles: why it is based on yes/no responses? Why wasn’t all the population described according to the job activity? E.g. 79% nurse, 20% technician, 10% midwife, etc?
Answer: Insertion of a new analysis to organize the text as recommended
- Table 1. “Weekly workload”: how these precise categories could be created if not included in the questionnaire? Or were the categories already defined in the questionnaire? Otherwise, I would have expected to have e.g. also someone with e.g. 32 or 38 hours/week
Answer: A footnote was inserted in the Table explaining how the working hours for nursing professionals in Brazil occur.
- Table 1: “Total monthly income”: and what about those with 10 MS/month?
Answer: Fixed entering the equal or greater than 10 MS/month sign.
- Table 02 and 03. Why 02 and 03 and not 2 and 3? Why “Source: The Author” in the footnote?
Answer: Correction was made regarding the numbering of tables. Excerpt: “Source: The Author” removed.
- Table 3. What is “a”? “ab”?2. Table 4. Formatting of the table issues.
Answer: Table 3 was excluded and the information was reorganized in Table 2.
Discussion
- Limitations section has to be expanded.
Answer: Redone and expanden section
Ref. lists
- Not in line with journal style.
Answer: All references have been reviewed and adapted to the journal's recommendations, as indicated in the section "Guidelines for Authors” - site International Journal of Environmental Research and Public Health

Reviewer 2 Report
COVID-19 and the mental health of nursing professionals in Brazil: The Correlation between social and clinical contexts and psychopathological symptoms
Comments to the authors
Thank you so much for the opportunity to review this valuable work! The work presented here is of importance and is quite compelling. However, there are some minor and major concerns that the authors could give consideration:
Abstract: This is well written and gives a summary of the paper. However, it may need tightening a bit more in the results and concluding statement. For example, when the authors state that “There was a lower mean (0.55, SD=0.45) in the south region and a higher mean (0.73, SD=0.50) in the north region when comparing….. It is not clear to the reader what that (higher/lower) means. And having a p-value for the subsequent variables in the abstract doesn’t help clarify the result and hence, the conclusions drawn. Please, consider tightening the abstract.
Introduction: Is well written but is quite brief. You may consider reviewing more literature to give a good foundation for the subsequent sections. For example, you could introduce some of the key concepts, the measurement scale, the rationale for the scale and measurement units in the introduction etc. This will allow the reader to have some introductory knowledge so that when they read the methods, results, and discussion sections, some of the concepts are already clarified in this section.
Materials and methods: The opening statement in this section is unclear to me. Requires some reworking/rephrasing: .... Cross-sectional study…
As mentioned above – the rationale for selecting SCL-40-R is not given and a clear description of what this scale measure is also missing. These will be important for the reader to make sense of the study finding.
Data analysis: the statistical tool adopted in the methods section of the paper does not march what is stated in the title. The authors talk about a correlation but in the methods section, they have a non-parametric analysis – Wilcoxon-Mann-Whitney and Kruskal-Wallis tests. There is no justification for the choice of these statistical methods of analysis. With a larger sample size, one may raise the question why these kind type of analysis. If the data violated the assumptions of normality, there is a need to indicate that to let the reader know. If that is the case, then this could be a limitation of the study that could be highlighted/pointed out so that caution is taken not to generalize the findings to the entire population of study.
The methods section requires some additional information to allow for replicability.
Results section: Females are overrepresented in the study – is this because nursing is a female-dominated profession? Or could it have been an omission in the design? Were efforts made to also represent the male voices within the data? What of the other races as it seems that white nurses were overly represented in your sample?
Table 03: It is unclear what the authors mean by the letters a, b, & ab. This makes it hard to interpret the results in the table.
There is also the mention of geographical differences. The results could benefit from disaggregating the results by region a little so as to provide regional-specific findings. This can allow for context-specific recommendations. Given that the authors do not tell us more about Brazil – no context information about the study sight, it is also hard to make sense of the regional differences. It might be helpful to give more information about the mental health situation in the different regions under study.
Giving more information about the methods will also be important in making sense of the table 02 & 04.
Discussion: The discussion has a lot more literature which is a positive thing. However, this section will benefit more from a detailed introduction with more literature that introduces some of the literature captured here. In line198 – the authors talk about “men and women are equally affected by mental health problems…..” Given that men are underrepresented in the study sample, I find this statement misleading or inaccurate unless there is evidence that the male sample in the study is representative of the population.
Limitations: The authors may consider putting forward a detailed limitation section as already capture above.
The conclusion section could also benefit from more policy-oriented recommendations as well as directions for future research.
Author Response
We, the authors of the article entitled “COVID-19 and the mental health of nursing professionals in Brazil: relationships between social and clinical contexts and psychopathological symptoms”, are grateful for the comments made on our article. We consider all the comments and suggestions of the referees and present our response to each one of them below. We reinforce that all suggestions were accepted (as highlighted below). Changes made to the text are highlighted in yellow.
(x) Moderate English changes required
Answer: Provided the complete revision of the text for the English language.
Abstract: This is well written and gives a summary of the paper. However, it may need tightening a bit more in the results and concluding statement. For example, when the authors state that “There was a lower mean (0.55, SD=0.45) in the south region and a higher mean (0.73, SD=0.50) in the north region when comparing….. It is not clear to the reader what that (higher/lower) means. And having a p-value for the subsequent variables in the abstract doesn’t help clarify the result and hence, the conclusions drawn. Please, consider tightening the abstract.
Answer: Abstract was fully restructured, presenting synthetically all the necessary elements and following the proposed recommendations and aiming to meet the recommendations of Reviewers 1 and 2.
Introduction: Is well written but is quite brief. You may consider reviewing more literature to give a good foundation for the subsequent sections. For example, you could introduce some of the key concepts, the measurement scale, the rationale for the scale and measurement units in the introduction etc. This will allow the reader to have some introductory knowledge so that when they read the methods, results, and discussion sections, some of the concepts are already clarified in this section.
Answer: The introduction was rewritten in order to clarify the concepts and understanding of the text.
Materials and methods: The opening statement in this section is unclear to me. Requires some reworking/rephrasing: .... Cross-sectional study…
Answer: Paragraph has been rewritten
As mentioned above – the rationale for selecting SCL-40-R is not given and a clear description of what this scale measure is also missing. These will be important for the reader to make sense of the study finding.
Answer: Accepted request. Provided more information about the instrument.
Data analysis: the statistical tool adopted in the methods section of the paper does not march what is stated in the title. The authors talk about a correlation but in the methods section, they have a non-parametric analysis – Wilcoxon-Mann-Whitney and Kruskal-Wallis tests. There is no justification for the choice of these statistical methods of analysis. With a larger sample size, one may raise the question why these kind type of analysis. If the data violated the assumptions of normality, there is a need to indicate that to let the reader know. If that is the case, then this could be a limitation of the study that could be highlighted/pointed out so that caution is taken not to generalize the findings to the entire population of study.
The methods section requires some additional information to allow for replicability.
Answer: The title was changed in order to make it consistent with the method of analysis used in the study. changes were made to the method in order to meet the suggestions.
Results section: Females are overrepresented in the study – is this because nursing is a female-dominated profession? Or could it have been an omission in the design? Were efforts made to also represent the male voices within the data? What of the other races as it seems that white nurses were overly represented in your sample?
Answer: In response to the questioning, a reference was introduced in the discussion that demonstrates the predominance of women in Brazilian nursing, in a proportion similar to that found in our study. This reference is a study carried out with 1,804,535 Brazilian nursing professionals, thus, it presents a reliable overview of their sociodemographic and work profile.
Table 03: It is unclear what the authors mean by the letters a, b, & ab. This makes it hard to interpret the results in the table.
Answer: Table 3 was excluded and the information was reorganized in Table 2 to make it more interpretable.
There is also the mention of geographical differences. The results could benefit from disaggregating the results by region a little so as to provide regional-specific findings. This can allow for context-specific recommendations. Given that the authors do not tell us more about Brazil – no context information about the study sight, it is also hard to make sense of the regional differences. It might be helpful to give more information about the mental health situation in the different regions under study.
Answer: A brief contextualization of the Brazilian regions in the method is presented.Giving more information about the methods will also be important in making sense of the table 02 & 04.
Answer: The method was rewritten in an attempt to meet this recommendation.
Discussion: The discussion has a lot more literature which is a positive thing. However, this section will benefit more from a detailed introduction with more literature that introduces some of the literature captured here. In line198 – the authors talk about “men and women are equally affected by mental health problems…..” Given that men are underrepresented in the study sample, I find this statement misleading or inaccurate unless there is evidence that the male sample in the study is representative of the population.
Answer: The intro has been restructured
Limitations: The authors may consider putting forward a detailed limitation section as already capture above.
Answer: Section redone and expanded.
A seção de conclusão também pode se beneficiar de recomendações mais orientadas para políticas, bem como orientações para pesquisas futuras.
Resposta: Seção refeita e ampliada.

Round 2
Reviewer 1 Report
Congratulations to the Authors as they were able to solve all the issues raised during the peer review process.